

# Relationships between locomotive and non-locomotive MVPA and '*ikigai*' in older Japanese adults

Soma Tsujishita[1], Masaki Nagamatsu[2], Aiko Imai[3] and Kiyoshi Sanada[4]

[1] Faculty of Rehabilitation Department of Physical Therapy, Kobe International University, Kobe, Hyogo, Japan
[2] Faculty of Research Organization of Science and Technology, Ritsumeikan University, Kusatsu, Shiga, Japan
[3] Faculty of Health Sciences, Suzuka University of Medical Sciences, Suzuka, Mie, Japan
[4] Faculty of College of Sport and Health Science, Ritsumeikan University, Kusatsu, Shiga, Japan

## ABSTRACT

**Objective:** This study aimed to investigate the relationship between physical activity (PA) of moderate to vigorous intensity (MVPA) and *ikigai* in older Japanese adults. We evaluated the amount of PA using an activity meter equipped with a three-axis accelerometer. The measured activities were classified into locomotive PA and non-locomotive PA to clarify not only the relationship between MVPA and *ikigai*, but the types of activities that lead to increased *ikigai* as well.

**Methods:** Participants were 86 community-dwelling older adult Japanese men and women. Measurement items included basic information (age, sex, BMI, and the number of underlying diseases), PA, Self-completed Occupational Performance Index (SOPI), and *ikigai* (the K-1 Scale). Confounding factors presumed to be related to ikigai were also elicited from previous studies. Associations of ikigai into three groups (high, middle, and low) with PA were analyzed by group comparisons and multivariate analyses.

**Results:** The comparison of PA indices among the high, middle, and low *ikigai* groups revealed that non-locomotive MVPA is significantly associated with *ikigai* ($p < 0.05$). In multiple comparisons, the low *ikigai* group was significantly and negatively affected by non-locomotive MVPA ($p < 0.05$) compared to the middle and high *ikigai* groups. Furthermore, in a multivariate analysis with *ikigai* as the dependent variable and sex, age, locomotive MVPA, and non-locomotive MVPA as independent variables, only non-locomotive MVPA ($p < 0.05$) was associated.

**Discussion:** These results suggest that non-locomotive MVPA is effective in enhancing *ikigai*.

Corresponding author
Soma Tsujishita,
tsujishita@kobe-kiu.ac.jp

## INTRODUCTION

Successful aging is a state in which people accept aging-related phenomena as normal changes, minimize morbid disorders, and achieve social and psychological satisfaction (*Rowe & Kahn, 1987*). In recent years, much attention has been paid to the concept of "*ikigai*" as an important indicator of successful aging (*Fukuzawa et al., 2019*). *Ikigai* is a

comprehensive concept that encompasses not only personal satisfaction and happiness but also social satisfaction (*Fukuzawa et al., 2019*). East Asians, including Japanese, tend to be intrinsically motivated when goals are established by their friends or family members on their behalf, whereas Westerners tend to be motivated by their own goal setting (*Iyengar & Lepper, 1999*). It has also been demonstrated that the achievement of each goal is related to well-being (*Oishi & Diener, 2001*). *Ikigai*, one of the indicators of well-being, is said to be the most used indicator of well-being in Japanese studies of older adults (*Yamamoto-Mitani & Wallhagen, 2002*). These suggest that there is a link between intrinsic motivation to achieve goals and *ikigai*. In other words, it is assumed that the concept of *ikigai* is important for the Japanese when considering the extension of healthy life expectancy.

The relationship between *ikigai* and health status has been investigated as it pertains to activities of daily living (ADL) impairment (*Mori et al., 2017*; *Tomioka, Kurumatani & Hosoi, 2016*; *Tomioka et al., 2015*), participation in care prevention and community activities (*Harada et al., 2021*; *Sasaki & Hirano, 2020*), health-related lifestyle habits (exercise, diet, sleep, and other habits) (*Kinoshita et al., 2020*), and subjective health (*Nakao et al., 2021*) among community-dwelling older adults. In our recent study, we found that overlapping physical, cognitive, and social frailties have adverse effects on *ikigai* among community-dwelling Japanese older adults (*Tsujishita, Nagamatsu & Sanada, 2022*). Thus, *ikigai* is important to maintain and improve health status.

On the other hand, physical activity (PA) is also important for extending healthy life expectancy. The World Health Organization (WHO) states that 150 to 300 min of moderate-intensity aerobic exercise or 75 to 150 min of high-intensity aerobic exercise, or a combination of both, of equivalent duration and intensity, should be performed per week (*World Health Organization (WHO), 2020*). Previous studies that investigated the association between PA and mortality reported a negative correlation between higher total PA of moderate to vigorous intensity (MVPA) and lower mortality (*Saint-Maurice et al., 2018*). It is also generally known that there are sex differences in PA. A previous study of older Australian adults reported that males were more active than females in terms of PA as assessed by the International Physical Activity Questionnaire for 60 min or more/per week (*Azevedo et al., 2007*). A previous study of elderly Japanese subjects reported that the percentage of adherence to "at least 150 minutes/week of moderate to vigorous intensity PA lasting at least 10 minutes" as measured by an accelerometer was 10.8% for men and 9.9% for women, showing no significant difference; however, when total PA was examined by sex, women were significantly more physically active (16.1 METs/hour/day) than men (14.0 METs/hour/day) (*Amagasa et al., 2021*). Therefore, the influence of gender differences should be taken into account when evaluating PA.

In addition, in recent years, In Japan, the Ministry of Health, Labour and Welfare (MHLW) recommends MVPA, which includes walking as well as various daily activities such as housework, from the perspective of lifestyle disease prevention (*Ministry of Health, Labour and Welfare of Japan, 2011*). It also points out the importance of dividing MVPA into two types: locomotive PA, which consists of walking and running, and non-locomotive PA, which focuses on daily activities such as washing clothes, washing dishes, moving small loads, and vacuuming (*Tanaka et al., 2013*). *Imai et al. (2020)*

reported that non-locomotive PA, consisting mainly of daily activities such as washing clothes, washing dishes, moving small loads, and vacuuming, compared to locomotive PA consisting of walking and running, may help prevent depressive symptoms in elderly Japanese women. Therefore, MVPA is thought to improve the health status of the elderly, and classifying MVPA into locomotive PA and non-locomotive PA will lead to the consideration of effective intervention methods.

However, we have not found any previous studies reporting an association between *ikigai* and MVPA. A reference in this regard is a previous study that investigated the relationship between quality of life (QOL) and MVPA.

WHO defines QOL as "an individual's perception of his or her life situation about goals, expectations, standards, or interests within the culture and values in which he or she lives" (*WHOQOL Group, 1995*). Also, in older adults, it has been reported that there is a close relationship between *ikigai* and QOL (*Tsuzishita & Wakui, 2021*). Regarding the relationship between QOL and the amount of PA, a systematic review of studies that investigated this relationship in subjects aged 18–65 years found strong evidence demonstrating that QOL is affected by PA for older adults (*Marquez et al., 2020*). One issue with many previous studies of older adults aged 65 and older is that the amount of PA was generally assessed by questionnaires, and few studies have investigated the relationship between QOL and objective quantification of PA such as that measured by accelerometers, for example. Indeed, only one study investigated the relationship between MVPA and QOL in older adults aged 65 years and older using accelerometers, reporting that increases in MVPA significantly improved QOL (*Awick et al., 2017*). These previous studies suggest that MVPA may be associated with *ikigai*.

This study aimed to investigate the relationship between MVPA and *ikigai* in older Japanese adults. We evaluated the amount of PA using an activity meter equipped with a three-axis accelerometer. The measured activities were classified into locomotive PA and non-locomotive PA to clarify not only the relationship between MVPA and *ikigai*, but the types of activities that lead to increased *ikigai* as well.

## MATERIALS AND METHODS

### Study design

Data were collected as previously described in *Tsujishita, Nagamatsu & Sanada (2022)*. Specifically, this cross-sectional study was conducted from June 2021 to November 2021. To recruit the target population, we asked the Regional Comprehensive Care Center in USA City, Oita Prefecture, to display posters and also to distribute advertisements to elderly residents in the area with their consent to cooperate. Participants were community-dwelling older adults aged 65 years and older who were verbally informed of the purpose and content of the study. Participants were informed that their participation was voluntary, that there would be no disadvantages even if they did not respond to the questionnaire, that the study could be terminated even after they consented to participate without any repercussion, and that they would not be identified because data would be processed anonymously. Individuals who agreed to participate in this study signed a consent form. This study was approved by the Ritsumeikan University Ethics Review

Committee for Medical Research Involving Human Subjects (Review No. BKC-LSMH-2021-011).

## Participants

Participant characteristics and exclusion criteria are described below. We recruited 86 community-dwelling older adults (21 males and 65 females; mean age ± standard deviation: 74.0 ± 6.2 years) living in USA City, Oita Prefecture, Japan, and obtained their consent. The survey was conducted from July 5, 2021, to November 30, 2021.
The exclusion criteria were as follows:

1) Those with a confirmed or suspected COVID-19 infection.
2) Those who had difficulty answering the questionnaire due to cognitive decline.
3) Those who were certified as requiring long-term care (required long-term care levels ≥1).
4) Those with a history of mental illness.
5) Those who had undergone orthopedic surgery or had movement restrictions.
6) Those deemed by a physician as ineligible to participate in the study due to illness.

Data from the 86 participants were subjected to analysis (Fig. 1).

## Sample size calculation

G*Power 3.1 software (Heinrich Heine University, Düsseldorf, Germany) was used to calculate the sample size for the Kruskal-Wallis test used for group comparisons of *ikigai* with a power of 80%, alpha error of 0.05, and effect size of 0.40 (large). The number of participants required for this study was determined to be 80. To account for possible dropouts, we recruited 86 participants.

## Measurement items

Measurement items included basic information (age, sex, height, weight, BMI, and the number of underlying diseases), PA, Self-completed Occupational Performance Index (SOPI), and *ikigai* (the K-1 Scale). Confounding factors presumed to be related to *ikigai* were also elicited from previous studies (*Harada et al., 2021*; *Sasaki & Hirano, 2020*; *Fukuzawa et al., 2019*; *Mori et al., 2017*; *Tomioka, Kurumatani & Hosoi, 2016*; *Tomioka et al., 2015*; *Roberson, Davies & Davidoff, 2000*).

Active Style Pro (HJA-350IT; Omron Corporation, kyoto, Japan), an activity meter equipped with a three-axis acceleration sensor, was used to evaluate the number of steps and PA. The device is 74 mm (Width) × 46 mm (Height) × 34 mm (Depth) and weighs 60 g. It is attached to the waist with a clip for measurement. Calculate the number of steps and the amount of activity based on three-axis acceleration data using a unique algorithm for each. The number of steps is counted as a walk when the amplitude of the acceleration waveform exceeds a predetermined threshold value and the motion continues for 2 s. In calculating the amount of activity, regardless of the activity intensity, the change in the gravity acceleration component of the acceleration signal is used to classify the activity into two categories: walking activities such as walking and running in which no upper body tilt

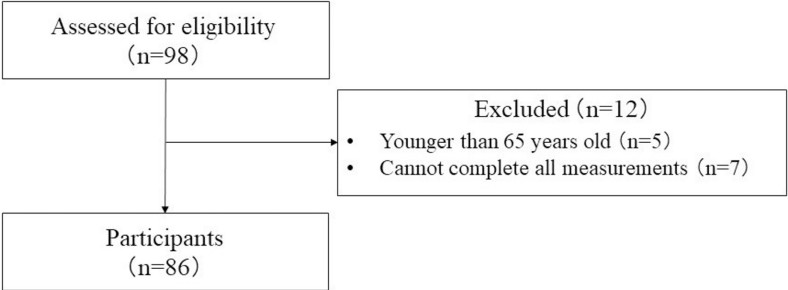

**Figure 1 Flowchart of study participant selection.**

change is observed during the activity, and daily activities such as carrying luggage and vacuuming in which upper body tilt change is observed during the activity. The unique feature of this system is that walking activity intensity and non-walking activity intensity is measured using the relationship equation between synthetic acceleration and activity intensity during each physical activity (*Ohkawara et al., 2011*; *Oshima et al., 2010*). Specifically, The intensity of PA was classified according to activity meter data as follows: sedentary behavior (1 to 1.5 metabolic equivalents (METs)), light-intensity PA (LPA: 1.6 to 2.9 METs), and MVPA (more than 3 METs). Additionally, each PA was classified into walking activity (locomotive PA) and daily living activity (non-locomotive PA). Participants were asked to wear the device on their waist from waking to bedtime, except when they were bathing or engaged in other in-water activities. The PA meter was worn for at least 10 h a day for at least 7 days in accordance with previous studies (*Kurita et al., 2017*; *Tanaka et al., 2013*).

The SOPI scores were used for qualitative interpretation of the physical activity. The SOPI is a nine-item self-administered questionnaire for measuring participation in activities of value to each individual. It consists of three questions on the three factors of leisure activity, productive activity, and self-care: work control, work balance, and work satisfaction. Each item is rated along a five-point scale (*Imai & Saito, 2010*). In this study, scores for leisure activity, productive activity, and self-care were used to evaluate the quality of ADL in addition to scores of ADL assessment. The authors have permission to use this instrument from the copyright holders.

The *Ikigai* scale for the elderly (the K-1 Scale) was used to assess *ikigai*. The K-1 Scale consists of 16 items and four sub-factors: "self-actualization and motivation," "sense of fulfillment," "will to live," and "sense of being." The questions are answered on a scale ranging from yes (2 points), neither (1 points), to no (0 points), and the total score is used to evaluate an individual's sense of purpose in life (*ikigai*) (*Kondo & Kamada, 2003*). The highest possible score is 32 (16 × 2), with a higher score indicating greater *ikigai*. There is no cutoff value for this scale. The authors have permission to use this instrument from the copyright holders.

Confounding factors included the presence or absence of an exercise habit (at least 2 days per week, average exercise time of at least 30 min), years of education (6–9 years, 10–13 years), work status, financial stability, marital status (married, bereaved/separated, never married), falls (in the past year), and hospitalization (in the past year), all of which

were considered to be associated with *ikigai* (*Harada et al., 2021*; *Nakao et al., 2021*; *Kinoshita et al., 2020*; *Sasaki & Hirano, 2020*; *Mori et al., 2017*; *Tomioka, Kurumatani & Hosoi, 2016*; *Tomioka et al., 2015*).

**Statistical analysis**

Before each statistical analysis, a Shapiro-Wilk test was performed to check for normality. The results showed a non-normal distribution of the *ikigai* scores. The K-1 Scale, which was used to assess *ikigai*, did not have a cutoff value, the analysis method was based on the distribution of *ikigai* scores, with quartiles calculated and classified into three groups: the "low group" for the first quartile, the "middle group" for the first quartile to the third quartile, and the "high group" for the third quartile and above. Specifically, the *ikigai* were classified into three groups: high (more than 30 points), medium (21 to 29 points), and low (0 to 20 points). Numerical values and scores of each assessment item were compared between the groups using the $\chi^2$ test, Kruskal-Wallis test, and Mann-Whitney U test after cross-tabulation. Based on the results of these tests, multinomial logistic regression analysis using the forced entry method was performed, with *ikigai* as the dependent variable and factors extracted from each evaluation as independent variables. Prior to the multinomial logistic regression analysis, correlations between items were examined using Spearman's rank correlation coefficients to avoid multicollinearity. SPSS version 27 for Windows (IBM, Armonk, NY, USA) was used for data analysis, with the statistical significance level set at 5%.

## RESULTS

Characteristics of the 86 participants are shown in Table 1. Mean age was 74.0 ± 6.2 years, 21 (24%) were male, and 65 (76%) were female. The high, middle, and low *ikigai* groups comprised 23 (27%), 38 (44%), and 25 (29%) participants, respectively (Table 1).

The comparison of PA indices between male and female participants revealed significantly higher non-locomotive LPA ($p = 0.0002$, $p < 0.001$, effect size (r) = 0.403), non-locomotive MVPA ($p = 0.003$, $p < 0.05$, effect size (r) = 0.322), total non-locomotive PA ($p = 0.002$, $p < 0.05$, effect size (r) = 0.326), and total activity ($p = 0.020$, $p < 0.05$, effect size (r) = 0.250) in females (Table 2).

The comparison of PA indices among the high, middle, and low *ikigai* groups revealed that non-locomotive MVPA is significantly associated with *ikigai* ($p = 0.029$, $p < 0.05$) (Table 3). In multiple comparisons, the low *ikigai* group had significantly lower non-locomotive MVPA ($p < 0.05$) than the middle and high *ikigai* groups. The proportion of participants who liked to be physically active was significantly higher in the high *ikigai* group than in the low *ikigai* group ($p = 0.023$, $p < 0.001$, effect size ($\varphi$) = 0.303).

The comparison of leisure activity, productive activity, and self-care among the three *ikigai* groups revealed that productive activity ($p = 0.038$, $p < 0.05$) and self-care ($p = 0.013$, $p < 0.05$) were significantly related to *ikigai* (Table 4). In multiple comparisons, the low *ikigai* group was significantly lower in productive activity and self-care than the high *ikigai* group. Spearman's rank correlation coefficients for leisure time activities, productive activities, and self-care and non-locomotive MVPA showed no correlation in either case.

**Table 1 Patient characteristics.**

| | |
|---|---|
| Age (years) | 74.0 ± 6.2 |
| BMI (kg/m²) | 23.4 ± 3.4 |
| Step counts (steps/day) | 6,044.8 ± 3,082.7 |
| SB (METs · h) | 4.7 ± 1.8 |
| Locomotive LPA (METs · h) | 1.5 ± 0.8 |
| Non-locomotive LPA (METs · h) | 5.9 ± 2.3 |
| Locomotive MVPA (METs · h) | 0.9 ± 0.8 |
| Non-locomotive MVPA (METs · h) | 2.3 ± 1.3 |
| Total locomotive PA (METs · h) | 2.3 ± 1.2 |
| Total non-locomotive PA (METs · h) | 8.3 ± 3.3 |
| Total activity (METs · h) | 10.6 ± 4.0 |
| Ikigai (points) | 24.7 ± 6.3 |
| Leisure activity (points) | 10.9 ± 3.0 |
| Productive activity (points) | 11.2 ± 2.6 |
| Self-care (points) | 12.2 ± 2.1 |
| Total score (points) | 70.3 ± 18.0 |
| *Ikigai* groups (persons) | High: 23, Middle: 38, Low: 25 |
| Number of underlying diseases (persons) | None: 35, One: 43, Two: 7, Three: 0, Four: 1 |
| Sex (persons) | Male: 21, Female: 65 |
| Years of education (persons) | 6–9 years: 4, 10–13 years: 82 |
| Financial comfort (persons) | Comfortable: 41, Not comfortable: 45 |
| Family (persons) | Living alone: 11, Living with someone: 75 |
| Work (persons) | Employed: 23, Not employed: 63 |
| Marriage (persons) | Married: 55, Bereaved/separated: 26, Never married: 5 |
| Exercise habits (persons) | Yes: 53, No: 33 |
| Falls (persons) | Yes: 20, No: 66 |
| Hospitalization (persons) | Yes: 17, No: 69 |
| Like to be physically active (persons) | Yes: 54, No: 32 |

**Note:**
Data are presented as mean ± standard deviation or number. BMI, body mass index; SB, sedentary behavior; LPA, light-intensity physical activity; MVPA, moderate to vigorous physical activity.

Finally, multinomial logistic regression analysis was conducted with *ikigai* as the dependent variable and age, sex, locomotive MVPA, and non-locomotive MVPA as independent variables. Sex was incorporated as a categorical variable and age, locomotive MVPA, and non-locomotive MVPA as continuous variables. The analysis of internal correlations before conducting the multivariate analysis revealed significant correlations among variables, suggesting that the results of the correlation matrix for the items that showed significant differences in between-group comparisons have low multicollinearity. However, the r-value did not exceed 0.8. Multivariate analysis showed that the high ikigai

**Table 2 Comparison of physical activity indices between male and female participants.**

| Evaluation item | Sex[1] | | p-value | p[2] | Effect size (r) |
|---|---|---|---|---|---|
| | Male (n = 21) | Female (n = 65) | | | |
| Step counts (steps/day) | 4,861.9 (3,317.0, 7,460.3) | 5,515.2 (4,204.0, 7,828.3) | 0.510 | | 0.071 |
| SB (METs · h) | 4.3 (3.6, 5.9) | 4.4 (3.6, 5.7) | 0.767 | | 0.032 |
| Locomotive LPA (METs · h) | 1.5 (0.9, 2.1) | 1.4 (1.0, 1.7) | 0.709 | | 0.040 |
| Non-locomotive LPA (METs · h) | 4.1 (3.1, 5.6) | 6.5 (4.7, 8.1) | 0.0002 | * | 0.403 |
| Locomotive MVPA (METs · h) | 0.8 (0.4, 1.2) | 0.6 (0.4, 1.2) | 0.510 | | 0.071 |
| Non-locomotive MVPA (METs · h) | 1.8 (1.1, 2.1) | 2.3 (1.8, 3.3) | 0.003 | * | 0.322 |
| Total locomotive PA (METs · h) | 2.4 (1.9, 3.3) | 2.0 (1.5, 2.9) | 0.162 | | 0.151 |
| Total non-locomotive PA (METs · h) | 6.0 (4.5, 7.2) | 8.9 (6.7, 11.2) | 0.002 | * | 0.326 |
| Total activity (METs · h) | 8.6 (7.3, 10.4) | 11.2 (8.6, 13.7) | 0.020 | * | 0.250 |

Notes:
Data are presented as median (interquartile range). SB, sedentary behavior; LPA, light-intensity physical activity; MVPA, moderate to vigorous physical activity.
[1] Comparisons between sex were made by the Mann-Whitney U test.
[2] *: $p < 0.05$.

**Table 3 Comparison of physical activity indices among high, middle, and low *ikigai* groups.**

| | Evaluation item | Ikigai[1] | | | p-value | p[2] | Multiple comparisons | Effect size (r, φ) |
|---|---|---|---|---|---|---|---|---|
| | | a. High *ikigai* group (n = 23) | b. Middle *ikigai* group (n = 38) | c. Low *ikigai* group (n = 25) | | | | |
| Physical activity index | Age | 73.0 (70.0, 78.0) | 74.0 (70.0, 80.5) | 71.0 (68.0, 77.7) | 0.280 | | | |
| | BMI | 24.0 (21.1, 25.0) | 22.6 (20.7, 24.2) | 23.5 (22.1, 25.9) | 0.246 | | | |
| | Number of underlying diseases | 1.0 (0.0, 1.0) | 1.0 (0.0, 1.0) | 1.0 (0.0, 1.0) | 0.842 | | | |
| | Step counts (steps/day) | 5,938.0 (4,716.0, 8,201.8) | 5,061.8 (2,946.3, 7,607.0) | 5,225.1 (4,204.0, 7,446.3) | 0.357 | | | |
| | SB (METs · h) | 4.1 (3.8, 5.4) | 4.6 (3.9, 6.1) | 3.7 (3.3, 5.1) | 0.237 | | | |
| | Locomotive LPA (METs · h) | 1.4 (0.9, 1.7) | 1.3 (1.1, 1.9) | 1.4 (0.8, 1.6) | 0.856 | | | |
| | Non-locomotive LPA (METs · h) | 5.8 (4.6, 7.4) | 6.4 (4.5, 7.7) | 5.3 (3.7, 6.4) | 0.152 | | | |
| | Locomotive MVPA (METs · h) | 0.8 (0.4, 1.3) | 0.6 (0.3, 1.1) | 0.6 (0.4, 1.3) | 0.290 | | | |
| | Non-locomotive MVPA (METs · h) | 2.3 (1.9, 3.4) | 2.2 (1.7, 3.2) | 1.9 (1.0, 2.2) | 0.029 | * | a > c, b > c | 0.275, 0.225 |
| | Total locomotive PA (METs · h) | 2.3 (1.7, 2.9) | 2.0 (1.5, 3.0) | 2.1 (1.3, 2.9) | 0.738 | | | |
| | Total non-locomotive PA (METs · h) | 8.1 (6.6, 11.0) | 8.4 (6.1, 10.8) | 7.0 (4.7, 9.2) | 0.088 | | | |
| | Total activity (METs · h) | 9.8 (8.9, 13.6) | 10.9 (8.3, 13.1) | 9.3 (6.5, 11.2) | 0.163 | | | |
| | Male (n (%)) | 5 (25.0) | 8 (40.0) | 7 (35.0) | 0.618 | | | |
| | Have an exercise habit (n (%)) | 15 (28.8) | 24 (46.2) | 13 (25.0) | 0.703 | | | |

(Continued)

| Evaluation item | Ikigai[1] | | | p-value | p[2] | Multiple comparisons | Effect size (r, φ) |
|---|---|---|---|---|---|---|---|
| | a. High *ikigai* group (n = 23) | b. Middle *ikigai* group (n = 38) | c. Low *ikigai* group (n = 25) | | | | |
| Have 10–13 years of education (n (%)) | 25 (32.1) | 33 (42.3) | 20 (25.6) | 0.337 | | | |
| Employed (n (%)) | 5 (22.7) | 11 (50.0) | 6 (27.3) | 0.615 | | | |
| Financially comfortable (n (%)) | 15 (40.5) | 17 (45.9) | 5 (13.5) | 0.032 | * | | 0.289 |
| Living alone (n (%)) | 0 (0.0) | 3 (30.0) | 7 (70.0) | 0.003 | *** | | 0.380 |
| Married (n (%)) | 18 (33.3) | 26 (48.1) | 10 (18.5) | 0.037 | * | | 0.352 |
| Have had a fall (n (%)) | 5 (26.3) | 9 (47.4) | 5 (26.3) | 0.873 | | | |
| Have been hospitalized (n (%)) | 5 (29.4) | 8 (47.1) | 4 (23.5) | 0.909 | | | |
| Like to be physically active (n (%)) | 19 (35.8) | 25 (47.2) | 9 (17.0) | 0.023 | *** | | 0.303 |

**Notes:**
Data are presented as median (interquartile range). BMI, body mass index; SB, sedentary behavior; LPA, light-intensity physical activity; MVPA, moderate to vigorous physical activity.
[1] Comparisons among the three Ikigai groups were performed by the Kruskal-Wallis test. Multiple comparisons were performed using the Mann-Whitney U test with Bonferroni's correction ($p < 0.05/3 = 0.017$) to account for multiplicity (with significant differences between groups). For the nominal scale, the $\chi^2$ test and Fisher's direct method were used.
[2] *: $p < 0.05$, ***: $p < 0.001$.

**Table 4  Comparison of SOPI scores among the high, medium, and low *ikigai* groups.**

| | Evaluation item | Ikigai[1] | | | p-value | p[2] | Multiple comparisons | Effect size (r, φ) |
|---|---|---|---|---|---|---|---|---|
| | | a. High *ikigai* group (n = 23) | b. Middle *ikigai* group (n = 38) | c. Low *ikigai* group (n = 25) | | | | |
| SOPI score | Leisure activity | 12.0 (11.0, 13.0) | 11.0 (9.0, 12.0) | 11.5 (6.0, 12.0) | 0.106 | | | |
| | Productive activity | 12.0 (12.0, 14.1) | 12.0 (9.0, 12.0) | 11.0 (9.0, 12.0) | 0.038 | * | a > c | 0.271 |
| | Self-care | 12.0 (12.0, 15.0) | 12.0 (12.0, 15.0) | 12.0 (9.0, 12.0) | 0.013 | * | a > c, b > c | 0.289 |

**Notes:**
Data are presented as median (interquartile range).
[1] Comparisons among the three ikigai groups were performed by the Kruskal-Wallis test. Multiple comparisons were performed using the Mann-Whitney U test with Bonferroni's correction ($p < 0.05/3 = 0.017$) to account for multiplicity (with significant differences between groups). For the nominal scale, the $\chi^2$ test and Fisher's direct method were used.
[2] *: $p < 0.05$.

group was only affected by non-locomotive MVPA (odds ratio: 2.484, 95% confidence interval [1.189–5.188], $p$-value = 0.015) (Table 5).

# DISCUSSION

The purpose of this study was to investigate the relationship between locomotive and non-locomotive MVPA and *ikigai* to explore effective interventions to enhance *ikigai* in older Japanese adults.

By sex, non-locomotive LPA, non-locomotive MVPA, total non-locomotive PA, and total activity were significantly higher in female participants than in male participants.

**Table 5 Multinomial logistic regression analysis on the association between *ikigai* and non-locomotive MVPA.**

| | | OR[2] | 95% CI[2] | *p*-value | *p*[4] | | | OR[2] | 95% CI[2] | *p*-value | *p*[4] |
|---|---|---|---|---|---|---|---|---|---|---|---|
| Dependent variable: | Sex[3] | 1.53 | [0.37–6.23] | 0.555 | | Dependent variable: | Sex[3] | 1.20 | [0.26–5.63] | 0.816 | |
| Middle *ikigai* group[1] | Age | 1.09 | [0.98–1.20] | 0.111 | | High *ikigai* group[1] | Age | 1.07 | [0.96–1.19] | 0.231 | |
| | Locomotive MVPA | 0.54 | [0.20–1.51] | 0.242 | | | Locomotive MVPA | 0.92 | [0.35–2.44] | 0.869 | |
| | Non-locomotive MVPA | 2.56 | [1.25–5.26] | 0.010 | * | | Non-locomotive MVPA | 2.48 | [1.19–5.19] | 0.015 | * |

Notes:
MVPA, moderate to vigorous physical activity.
[1] The reference category for the dependent variable is the low ikigai group.
[2] OR, odds ratio; 95% CI, 95% confidence interval.
[3] The reference category for sex is male.
[4] *: $p < 0.05$.

The comparison of PA by *ikigai* (*i.e.*, high, middle, and low) revealed a significant correlation between non-locomotive MVPA and *ikigai*. Moreover, the comparison of leisure activity, productive activity, and self-care revealed a significant relationship between *ikigai* and productive activity, and self-care. Moreover, the proportion of participants who liked to be physically active was higher in the high *ikigai* group than in the low *ikigai* group, suggesting that participants with high *ikigai* are more likely to be physically active than those with low *ikigai*. Furthermore, the multivariate analysis using *ikigai* as the dependent variable and age, sex, locomotive MVPA, and non-locomotive MVPA as independent variables revealed that only non-locomotive MVPA was associated with high *ikigai*.

The comparison of PA indices by *ikigai* (high, middle, and low) revealed a significant relationship between *ikigai* and non-locomotive MVPA. Regarding the relationship between *ikigai* and PA, previous studies have reported an association between high *ikigai* and the amount of PA (*Belice et al., 2022*; *Okuzono et al., 2022*; *Kabasawa et al., 2021*; *Hirooka et al., 2021*). However, previous studies used questionnaires to evaluate PA and did not use tri-axial accelerometers to calculate objective values as in this study. Another unique aspect of this study is that it not only measured PA but also analyzed *ikigai* and physical activity by classifying PA into two types: locomotive PA and non-locomotive PA. As mentioned earlier, the concept of QOL is similar to that of *ikigai*. Regarding the relationship between QOL and PA, some studies use triaxial accelerometers for PA. A systematic review of studies on the relationship between QOL and the amount of PA in subjects aged 18–65 years found strong evidence demonstrating that PA affects QOL for older adults (*Marquez et al., 2020*). Most previous studies measured PA using questionnaire responses to examine its relationship with QOL, and few studies objectively quantified PA using accelerometers or other instruments. For example, only one study investigated the relationship between MVPA and QOL in older adults using accelerometers. An increase in MVPA has been reported to significantly increase QOL (*Awick et al., 2017*). This association is valid at least to some extent given the strong relationship between QOL and *ikigai*, and although MVPA was not classified as locomotive or non-locomotive, it is suggested that QOL is associated with non-locomotive MVPA. These previous studies suggest that there is some validity to the association between *ikigai* and non-locomotive MVPA, but the mechanism is unknown. As an idea to

suggest this, previous research on the K-1 scale, the measure of *ikigai* in this study, may be helpful (*Kondo & Kamada, 2003*). In the study, a K-1 Scale was created and an operational definition of *ikigai* was developed based on the results of the K-1 Scale. The definition is reported as follows "*Ikigai* is a sense of purpose and motivation in one's daily life, and a sense of commitment to life with the awareness that one is useful to one's family and others and that one is indispensable to them. It is also a sense of pride when one feels that one has achieved something, improved in some small way, or has been recognized by others." Considering these facts from the results of this study, it is likely that activities that have purpose and meaning are more likely to be imagined as activities of daily living that involve physical activity, such as housework or hobbies, rather than activities that do not involve much physical activity, such as walking. Thus, *ikigai* may have been associated with non-locomotive MVPA. However, this is only speculation, and future studies should examine in detail what activities are associated with *ikigai* in non-locomotive MVPA.

The present study found significant associations between *ikigai* and productive activity, and self-care. In a previous study targeting 618 older adults, 79.0% of men and 87.4% of women answered "yes" to the question "Do you feel fulfilled when you are engaged in leisure activities?" (*Harada et al., 2011*). According to this survey, roughly 80% of both men and women felt fulfilled during their leisure time activities. In the present study, *ikigai* was associated with productive activity (*e.g.*, housework, community volunteer work) and self-care (*e.g.*, bathing, shopping). No association was found between *ikigai* and leisure activity (*e.g.*, hobbies). This finding differed from previous reports, possibly because the present study compared satisfaction with leisure time activity, productive activity, and self-care among different *ikigai* groups, whereas previous studies used questions such as "Do you feel a sense of purpose in life when you are engaged in leisure activities?" and solicited yes/no answers. Our findings suggest that older adults are more likely to find fulfillment in familiar activities such as productive activities and self-care, rather than free activities such as leisure time activities. In addition, more participants in the high *ikigai* group responded that they liked to be physically active compared with the low *ikigai* group. In the Cabinet Office's survey on the attitudes of older adults regarding their daily lives, the most common response to the question, "When do you feel motivated to live?" was during "hobbies and sports" (47.3%), as they found activities involving more physical exercise to be worthwhile (*Cabinet Office of Japan, 2015*). Thus, previous studies as well as the present study suggest the need for efforts to encourage older people to enjoy PA to enhance their *ikigai*.

Next, female participants had significantly higher non-locomotive LPA, non-locomotive MVPA, total non-locomotive PA, and total activity than their male counterparts, suggesting that females are more physically active overall. Sex-dependent differences in the amount of PA vary by country and are presumed to reflect the culture of each country (*Amagasa et al., 2021*; *Azevedo et al., 2007*). Our finding that women are significantly more physically active than men is consistent with a previous report in Japan. The difference from previous studies is that in this study, daily activity is characterized as being higher in women than in men. The reason for this, it is difficult to clarify in this study. A previous study by the Ministry of Internal Affairs and Communications in Japan

classified the total time for each daily activity in the lives of elderly people into the following three categories: "primary activities" (sleeping, eating, and personal errands, *i.e.*, basic activities of living), "secondary activities" (work and housework), and "tertiary activities" (activities that are not primary or secondary activities, such as leisure activities) (*Statistics Bureau of the Ministry of Internal Affairs and Communications of Japan, 2017*). The time spent performing each category of activities by older adults (aged ≥60 years) was examined, and women were found to spend more time on secondary activities than men (*Statistics Bureau of the Ministry of Internal Affairs and Communications of Japan, 2017*). Specifically, among primary activities, men spent more time "sleeping" than women, and women spent more time "doing personal errands" than men; among secondary activities, men spent the most time on "work," whereas women spent the most time on "housework." Of the tertiary activities, men spent more time on "TV/radio/newspapers/magazines," "rest/relaxation," "hobbies/entertainment," and "volunteer activities/social participation" than women, and women spent more time "socializing/associating" than men (*Statistics Bureau of the Ministry of Internal Affairs and Communications of Japan, 2017*). Thus, sex-dependent differences in lifestyle in Japanese people may explain differences in the amount of PA in old age. For example, women spend more time fulfilling domestic roles, whereas men spend more time working outside the home. In the present study, non-locomotive LPA, non-locomotive MVPA, total non-locomotive PA, and total activity were higher in women than in men, possibly due to the unique differences in lifestyle between Japanese men and women in old age.

Finally, the multivariate analysis revealed that only non-locomotive MVPA was associated with high *ikigai* after controlling for age, sex, and locomotive MVPA. Taken together, these findings suggest that non-locomotive MVPA is important in enhancing *ikigai*. A specific example of interventions may involve asking older people to select non-locomotive MVPA of 3 METs or more (*e.g.*, playing outside with grandchildren, engaging in volunteer activities such as weeding or picking up trash in the city) to incorporate into their daily lives as the first step.

There are some limitations worth noting in this study. First, the survey period was from April to December 2021, and the possible effects of the COVID-19 pandemic, such as restrictions on going out and socializing, might have negatively affected *ikigai* and PA, resulting in low overall figures. Second, since the sample size was small, a future study with an increased sample size is warranted to examine further the relationship between PA level and the sense of fulfillment in life. In addition, we plan to utilize the results of this study to verify whether the aforementioned intervention method (*i.e.*, to have elderly people living in the community to choose and engage in a non-ambulatory activity of 3 METs or more in their daily lives) would increase their sense of fulfillment in life.

## CONCLUSIONS

The present study found that non-locomotive MVPA is significantly associated with *ikigai*, and that this type of PA may offer effective methods to increase *ikigai* in older people. Furthermore, the multivariate analysis with variables including age and sex revealed a

strong association between *ikigai* and non-locomotive MVPA. These results suggest that high non-locomotive MVPA is important for high *ikigai*.

## ACKNOWLEDGEMENTS

The authors would like to thank all subjects who participated in this study.

### Funding

The authors received no funding for this work.

### Competing Interests

The authors declare that they have no competing interests.

### Author Contributions

- Soma Tsujishita conceived and designed the experiments, performed the experiments, analyzed the data, prepared figures and/or tables, authored or reviewed drafts of the article, and approved the final draft.
- Masaki Nagamatsu conceived and designed the experiments, performed the experiments, authored or reviewed drafts of the article, and approved the final draft.
- Aiko Imai conceived and designed the experiments, analyzed the data, authored or reviewed drafts of the article, and approved the final draft.
- Kiyoshi Sanada conceived and designed the experiments, performed the experiments, analyzed the data, prepared figures and/or tables, authored or reviewed drafts of the article, and approved the final draft.

### Human Ethics

The following information was supplied relating to ethical approvals (*i.e.*, approving body and any reference numbers):

Ritsumeikan University Ethics Review Committee for Medical Research Involving Human Subjects

### Data Availability

The raw data is available in the Supplemental File.

### Supplemental Information

Supplemental information for this article can be found online at http://dx.doi.org/10.7717/peerj.15413#supplemental-information.

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
