# Peer review of "Relationships between locomotive and non-locomotive MVPA and ‘ikigai’ in older Japanese adults"

_PeerJ, doi:10.7717/peerj.15413_

## Round 0.1 · original submission · Major Revisions

Three authors provide many significant questions. You need to read it carefully and make a good revision. When addressing these questions please highlight the change where you revise in the manuscript. Also, prepare a rebuttal letter to answer these questions one by one.

Reviewer 1 ·

Basic reporting

Comment: L46-48:The sentences in 46-48 and the contents of the cited literature do not match, but from what perspective is the quotation?

Comment: L48: "There is also a link between intrinsic motivation and ikigai. "Is there a thesis to support this statement?

Comment: L57-59: The authors write, " In our recent study, we found that overlapping physical, cognitive, and social frailties have adverse effects on ikigai among community-dwelling Japanese older adults.". Is there a published paper with this result?

Comment: L77-78: Ref. Shimozuma, 2015 is a review paper. So please add the original article(s).

Comment: L81-84: There are few papers on physical activity and quality of life in the elderly using accelerometers, but there may be more studies in other age groups. Therefore, including the target age group in the text might be better.

Comment: L144-147: The sentences and the contents of the cited literature do not match.

Comment: L151-155: The sentences and the contents of the cited literature do not match.

Comment: L159-163: From which references do the authors summarize the confounding factors evidence?

Comment: L166: Please explain the criteria for dividing ikikai into three groups.

Comment: L283-285: The information in the text is not found in the cited literature. Are the authors citing the literature incorrectly?

Comment: Since many of the cited references are to papers written in Japanese, it would be kind to include a brief mention in the references section that the paper was written in Japanese.

Comment: The numbers shown in Table 1 (SB to Total activity) do not match the calculations from the raw data. Please check again.

Comment: In Table 2, the "1) The Mann-Whitney U test was performed to assess cognitive frailty." is unintelligible.

Comment: In Table 2, for which items did the authors use the X2 test?

Comment: In Table 3, what do the authors define for ***?

Experimental design

no comment

Validity of the findings

no comment

Reviewer 2 ·

Basic reporting

Thank you for the opportunity to review this article. Although the study focuses on an important research issue, the methodology used to achieve its objective may not be appropriate.

1. The purpose of this study was to investigate the relationship between locomotive and nonlocomotive MVPA and ikigai. However, the authors have presented MVPA comparison between the ikigai groups as the main result in the abstract. This only indicates the characteristics of MVPA between the groups.

2. Why did the authors not use both nonlocomotive and locomotive MVPA as independent variables in the multinominal logistic regression analysis? I think this analysis is necessary to achieve the study objective.

3. The authors calculated the sample size, but it is not clearly stated that the sample size is for which analysis.

4. How was each independent variable treated in the logistic model? Was each independent variable incorporated into the model as a continuous or categorical variable?

5. It is not possible to mention that “the low ikigai group was significantly and negatively affected by non-locomotive MVPA” (line 188) based on the results presented in Table 3. This table only shows that nonlocomotive MVPA in the low ikigai group was significantly lower than that in other groups.

6. The reason/mechanism underlying the observed association between nonlocomotive MVPA and ikigai was not sufficiently described in the Discussion section.

7. In the Conclusion section, the causal relationships between nonlocomotive MVPA and ikigai should not be mentioned due to the cross-sectional study design.

8. If the authors want to mention “non-locomotive MVPA is effective for enhancing ikigai, with activities that increase the housework of PA, such as productive and self-care activities, being particularly important,” (line 328) the relationship between nonlocomotive MVPA and these activities should be evaluated.

Experimental design

as stated above

Validity of the findings

as stated above

Reviewer 3 ·

Basic reporting

I have provided suggestions for additional citations and literature in the general comments. There are some results presented for which rationales are not clearly defined in the introduction (focus on sex differences, types of activities). Additional comments are in the general comments section.

Experimental design

The research question is defined, but the knowledge gap is not clearly identified. The study presents results for additional research questions not clearly defined. There are some areas in which the methods could be more clear, including how participants were recruited, how locomotive and non-locomotive PA was measured, and the ways in which groups were split into high/medium/low ikigai scores.

Validity of the findings

The analytic approach is not clearly justified (splitting ikigai groups).

Additional comments

1. The [Background] section of the abstract does not provide background – it provides a purpose and begins the methods of the current study. Either consider renaming the section to be [Objective] or reorganize the abstract to include background information.

2. Locomotive and non-locomotive PA are not terms that I am familiar with. I wonder if a brief definition or distinction of the differences of these types of PA in the abstract would be helpful to the reader.

3. The definition of ikigai provided in the introduction, line 44 (personal satisfaction and happiness and social satisfaction) is somewhat different from what I’ve seen in other papers. Ikigai has been described as analogous to a Western sense of purpose. The next couple of lines refer to goal setting and intrinsic motivation, which is more consistent with what purpose is and my prior understanding of what ikigai is. I wonder if the authors can expand upon the definition provided in line 44 and more clearly explain the connection to intrinsic motivation and goal setting.

4. Introduction, line 65: I believe more recent PA guidelines have removed the 10 min/bout requirement and suggested that any accumulation of PA is acceptable. I see the reference is from 2010 and would encourage the authors to examine more recent guidelines.

5. Line 66 – the correlation between MVPA and all-cause mortality – describe the direction of this association.

6. Line 70 – Again, it would be helpful to clearly define the distinction between non-locomotive and locomotive PA for the reader.

7. Introduction – It is not clear what the gap in the literature is that would describe the need for this study. Are the authors suggesting that ikigai is a component of quality of life? This needs to be explained more clearly.

8. Line 108 – It is not clear how participants were identified or recruited for participation.

9. It is not clear how the accelerometer data would be able to identify non-locomotive v. locomotive PA. The accelerometer is unable to capture the type of activity.

10. Line 165 – Statistical Analysis. It is not clear why the authors would divide ikigai into three groups. The scale is continuous and there are not predefined cut-points described in the manuscript. Dividing a continuous scale into a categorical one loses variability. If this change is made, the authors could use multiple linear regression to predict ikigai instead of multinomial logistic regression.

11. Line 213: Instead of starting the discussion with the sex and MVPA relationship, the authors should start by discussing the findings of the main hypothesis (MVPA and ikigai).

12. Line 191, Line 216, Line 283: instead of referring to the name of the scale (SOPI), refer to the construct that is being measured (valued activities). This improves the clarity for the reader.

13. In general, I’m not clear on why there is such a strong emphasis on sex differences in the results and discussion. If this is a goal of the paper, this rationale should be described more clearly in the introduction.

14. Line 263: The authors claim that no previous study has reported the relationship between ikigai and PA. This statement is incorrect. Here are a handful of studies that have examined ikigai and physical activity – there are likely more.

Kabasawa, K., Tanaka, J., Ito, Y., Yoshida, K., Kitamura, K., Tsugane, S., ... & Narita, I. (2021). Associations of physical activity in rural life with happiness and ikigai: a cross-sectional study. Humanities and Social Sciences Communications, 8(1), 1-10.
Belice, T., Ozkan Yildirim, N., Gursoy, U., Güleryüz, O., Demir, I., & Yüksel, A. (2022). The Relationship Between Physical Activity and Polypharmacy with Ikigai in a Population. EUREKA: Health Sciences, 2, 3-9.
Hirooka, N., Kusano, T., Kinoshita, S., Aoyagi, R., & Hidetomo, N. (2021). Association between healthy lifestyle practices and life purpose among a highly health-literate cohort: a cross-sectional study. BMC Public Health, 21(1), 1-8.
Okuzono, S. S., Shiba, K., Kim, E. S., Shirai, K., Kondo, N., Fujiwara, T., ... & VanderWeele, T. J. (2022). Ikigai and subsequent health and wellbeing among Japanese older adults: Longitudinal outcome-wide analysis. The Lancet Regional Health-Western Pacific, 21, 100391.

15. I know that ikigai is broader than the Western sense of purpose, but they are similar to some extent. There is a large literature examining the relationship between purpose and physical activity and referring to some of that in the discussion could be useful.

---

## Round 0.2 · Minor Revisions

Three reviewers still have several questions. Please revise it according to their comments/suggestions. Then, I will decide whether this paper is suitable for publication.

Reviewer 1 ·

Basic reporting

Comment: In Table 4, what do the authors define for ***?

Experimental design

no comment

Validity of the findings

no comment

Reviewer 2 ·

Basic reporting

Compared to the first manuscript, the revised manuscript seems to be even more unclear as the purpose, methods, results, and conclusions of the study did not correspond with each other.

Possible reasons are as follows:
1. Too many variables are used beyond the study objectives.
→ e.g., Although the purpose of this study was to examine the relationship between locomotive/nonlocomotive MVPA and ikigai, the authors have mentioned the importance of productive and self-care activities and a love of physical activity in the Conclusion section.

2. The justification for using each variable is unclear.
→ e.g., I understood that the SOPI scores were used for qualitative interpretation of locomotive activity, but it does not seem to be so according to your response (#7).

3. The interpretation of analysis results seems to be incorrect.
→ e.g., As the reviewer pointed out in the first manuscript, the comparison of variables by the ikigai groups was showed only the group characteristics. Thus, it is also not possible to mention that “the low ikigai group significantly and negatively affected productive activity and self-care compared to the high ikigai group” (Line 236) based on the results presented in Table 4.

Experimental design

as stated above

Validity of the findings

as stated above

Reviewer 3 ·

Basic reporting

No comment

Experimental design

No comment

Validity of the findings

No Comment

Additional comments

1. Thank you for providing definitions of locomotive and non-locomotive PA. It is still not clear to me how the accelerometer is able to determine walking and non-walking intensity based on the description provided. Generally, tri-axial accelerometers only capture movement on three planes and are not able to distinguish the type of activity.
2. Thank you for providing the cut-points for the three ikigai groups. Again, it is not clear why the authors chose to use arbitrary cut-points to categorize ikigai rather than using a continuous measure. The continuous measure would allow for more variability in ikigai and allow the authors to use less complex analyses (multiple linear regression v. multinomial logistic regression).
3. Line 207: “the "low group" for the first quartile to the first quartile” – This could be revised to say scores in the first quartile were put in the “low group”

---

## Round 0.3 · accepted · Accept

Congratulations on your achievement. I believe physical activity and Ikigai are very important for aged people. Your paper not only provides knowledge of why locomotor and non-locomotor PA are related to Ikigai but also their intercorrelations. I hope you keep working on the issue of physical activity and health in the future.